# Determinants Affecting Purchase Willingness of Contractors towards Construction and Demolition Waste Recycling Products: An Empirical Study in Shenzhen, China

**DOI:** 10.3390/ijerph18094412

**Published:** 2021-04-21

**Authors:** Bo Yu, Jiayuan Wang, Ying Liao, Huanyu Wu, Aslan B. Wong

**Affiliations:** 1College of Civil and Transportation Engineering, Shenzhen University, Shenzhen 518000, China; yuboszu@gmail.com (B.Y.); wuhuanyu@szu.edu.cn (H.W.); 2Department of Architecture and Civil Engineering, City University of Hong Kong, Hong Kong 999077, China; yingliao7-c@my.cityu.edu.hk; 3College of Computer Science and Software Engineering, Shenzhen University, Shenzhen 518000, China; aslan@szu.edu.cn

**Keywords:** construction and demolition waste, recycling product, purchase willingness, critical factor, contractor

## Abstract

Waste recycling is a critical method to effectively address environmental issues raised by construction and demolition waste (C&DW). As highlighted in previous studies, the contractor is considered the primary purchaser for recycled C&DW products. However, there is a limited understanding of the factors affecting the contractor’s purchase willingness towards C&DW recycling products. This study investigated these key drivers using the Exploratory Sequential Mixed Approach. Firstly, a hypothetical model was developed based on the theory of planned behavior (TPB). Secondly, a questionnaire survey was then employed in data collection. Thirdly, structural equation modeling (SEM) was adopted for data analysis. It is revealed that multiple factors directly affect the contractor’s purchase willingness towards C&DW recycling products. These factors include government measures, the contractor’s attitude, perceived behavioral control, perceived consumer effectiveness, and subjective norms. Besides, recycling product information indirectly affects the contractor’s purchase willingness. Based on the findings, the study provides strategies for the government, contractors, recycling enterprises, and public buyers to increase C&DW recycling products’ purchase willingness. Findings derived from the empirical study can be used as a theoretical reference for government departments to develop related promotion policies. Moreover, the suggestions provided are helpful to guide recycling enterprises to promote their products.

## 1. Introduction

Urbanization and urban regeneration have contributed substantial quantities of C&DW [1]. One example is the massive amount of waste produced in Shenzhen, China, which reached approximately 50 million tons from 2010 to 2017 [2]. Sound management of the waste is essential to tackle environmental and social issues during the development of a sustainable city [3,4,5]. As eliminating all the waste is unrealistic, reuse and recycling are critical methods to reduce the landfill rate of C&DW [6]. Remarkably, the recycling of C&DW is generally considered to be a sustainable and effective way to solve the problems raised by landfilling waste [7].

C&DW could be recycled into products such as bricks, mortar, aggregate, and lightweight wallboard [8,9,10,11,12]. These products are generally used in housing construction, municipal public engineering, etc. For example, recycled bricks can be used in municipal roads, parks, plazas, airports, and docks [13]. Recycled mortar is used in various masonry and plastering projects, and recycled aggregates are used in subgrade or building foundation cushions [14,15,16]. However, C&DW recycled products are habitually labeled as toxic, low quality, and cheap. As a result, the public is reluctant to accept and purchase such products compared to non-recycled products [17]. Thus, promoting the adoption of C&DW recycled products has raised significant interest among government and industry practitioners [13].

Human factors have recently become an increasingly hot topic in the field of C&DW management because stakeholders’ attitudes, willingness, and behavior could remarkably influence C&DW management practices. For instance, several studies have focused on determining contractors’ attitudes and behavior toward C&DW management [18,19]; There are the number of scholars focusing on designers’ attitude and behavior towards the minimization of C&DW [20] and the willingness to minimize C&DW [21]. Some studies emphasize the attitude, willingness, or behavior of other stakeholders, such as construction workers, project managers, government officials, environment consultants, and clients [22,23,24,25,26]. However, there is limited research, if any, focusing on the public’s willingness to purchase recycled C&DW products. As a result, the government and industry practitioners have no information to develop available strategies to promote C&DW recycled products. In addition, contractors are the primary purchasers of recycled C&DW products because they often need to be responsible for the procurement of materials needed for project construction. These materials (such as bricks, mortar, aggregate, etc.) could be derived from C&DW recycled products. Therefore, understanding the contractor’s purchase willingness towards recycled C&DW products will increase the recycling rate.

Hence, the study aims to explore the determinants affecting contractors’ purchase willingness for C&DW recycling products and analyze their interactions. The study also proposes a series of strategies to promote the purchase of recycled C&DW products for various stakeholders, such as the government, contractors, recycling enterprises, and public buyers. To be specific, there are four main research objectives: (1) To identify the main factors affecting the purchase willingness of contractors towards recycled C&DW products; (2) To construct a hypothetical model according to the relationships among these influencing factors; (3) To validate the hypothetical TPB model, and to reveal critical factors and paths affecting purchase willingness of contractors towards recycled C&DW products; (4) To develop practical recommendations that promote the use of recycled C&DW products by contractors based on the validation results of this empirical study. 

The remainder of the paper is organized as follows: Section 2 illustrates the theoretical basis and research hypotheses. Research methods and process are then presented in Section 3. The findings of this study are reported in Section 4. In-depth discussions can be found in Section 5. The final section sheds light on our conclusions and the study limitations as well as further research directions. 

## 2. Theoretical Basis and Research Hypotheses

### 2.1. Theory of Planned Behavior

In C&DW management research, many scholars have studied the factors that affect stakeholders’ behavior, attitudes, and willingness to minimize, decrease and recycle C&DW [18,24,26]. Psychological models are employed as a theoretical basis, such as the TPB.

The TPB, a theory developed by Ajzen, creates a linkage between individual beliefs and behavior [27]. This theory established the foundation to explore the effects of factors influencing behavioral choices [28]. TPB argues that behavior depends on willingness and perceived behavioral control, while willingness is constrained by attitude, perceived behavioral control, and subjective norm. Attitude is related to a person’s positive or negative evaluation of a particular behavior. Subjective norms denote the influence of external social factors associated with personal behavior. It is worth noting that attitude concentrates on people’s moral responsibilities or emotional feelings, while subjective norms focus on pressures and opinions from their surroundings [29]. Perceived behavioral control involves people’s perception on the convenience of performing a behavior. The TPB framework is suitable for studying individual’s willingness and behavior [30]. However, as it is difficult to track the actual behavior [29], this study explores the determinants that affect the contractor’s purchase willingness towards recycled C&DW products.

The TPB has been widely exploited in C&DW management. For example, Li et al. revealed that attitude and perceived behavioral control could remarkably affect the designers’ C&DW minimization behavior [20]. The empirical results of Li et al. show that subjective norms have a more significant effect on contractor employees’ C&DW reduction behavior than attitude and perceived behavioral control [18]. Yuan et al. also explored the motivational factors that determine project managers’ willingness towards C&DW reduction. The attitude was shown to be the most important influencing factor, followed by the subjective norms and perceived behavioral control [26]. Coincidentally, Wang et al. also declared that attitude is at play in affecting the design unit’s willingness to minimize C&DW [21]. These studies further demonstrate the TPB framework, viz., willingness is affected by attitudes, subjective norms, and perceived behavioral control. Therefore, hypotheses 1–3 are proposed: 

**Hypothesis 1** **(H1).***Attitude has a direct positive effect on contractors’ purchase willingness towards C&DW recycled products*.

**Hypothesis 2** **(H2).***Subjective norms have a direct positive effect on contractors’ purchase willingness towards C&DW recycled products*.

**Hypothesis 3** **(H3).***Perceived behavioral control has a direct positive effect on contractors’ purchase willingness towards C&DW recycled products*.

Additional explanatory factors can be incorporated into the TPB model as long as they deliver explanations of behavior or willingness [27]. For example, Wu et al. attempted to merge governmental supervision, project constraints, and economic viability into the TPB model [19]. The results demonstrated that these elements were more prominent than the initial determinants. Hence, this study incorporates other influencing factors, such as perceived consumer effectiveness, recycling product information, and government measures, that may affect contractors’ purchase willingness towards recycled C&DW products.

### 2.2. Perceived Consumer Effectiveness

Perceived consumer effectiveness stands for the degree to which customers solve environmental problems [31,32]. In general, people who hold a positive attitude toward green consumption are likely to prioritize green consumption behaviors when they perceive that their choices are conducive to improving environmental quality [33,34]. The empirical results of Roberts show that perceived consumer effectiveness could explain 33% of environmental consumption behavior [35]. Besides, Kabaday et al. revealed that this factor is the paramount determinant of green purchase willingness [36]. Therefore, perceived consumer effectiveness might also affect contractors’ purchase willingness towards recycled C&DW products. Hypothesis 4 was established as follows:

**Hypothesis 4** **(H4).***Perceived consumer effectiveness has a direct positive effect on contractors’ purchase willingness towards recycled C&DW products*.

### 2.3. Recycled Product Information

Product attributes (such as quality, brand, packaging), price, and sales channels are collectively referred to as product information [37]. Chang and Wildt mentioned that product information is a critical driver affecting potential consumers for product evaluation and actual purchases [38]. Gupta et al. also revealed that almost all consumers choose to remanufacture products rather than purchasing new ones because of price, brand, and quality [39]. Besides, Ares et al. emphasized the importance of price in consumers’ purchase decisions [40]. Krishnan et al. also concluded that although green product demand continues to increase, price is still the most important determinant for consumers [41]. Therefore, product information is considered one of the critical factors influencing recycled C&DW products’ purchase willingness. Hence, Hypothesis 5 was proposed: 

**Hypothesis 5** **(H5).***Recycling product information has a direct positive effect on contractors’ purchase willingness towards recycled C&DW products*.

### 2.4. Government Measures

Formulating incentive policies, creating demonstration projects, and increasing publicity are often considered government measures. Previous studies have specified that incentive policies could promote eco-friendly products [42]; government propaganda can influence mass behavior [43]. Liu et al. found through the implementation of demonstration projects financed by the government for renewable energy application buildings, the whole market, including both demand and supply sides, has been activated, and applications using renewable energy in buildings has been popularized effectively in China [44]. It is speculated that government measures will affect mainly the willingness to purchase recycled C&DW products. Therefore, Hypothesis 6 was the following proposed:

**Hypothesis 6** **(H6).***Government measures have a direct positive effect on contractors’ purchase willingness towards recycled C&DW products*.

Among the above six potential influencing factors, recycling product information and government measures are external factors that affect contractors’ purchase willingness towards recycled C&DW products, while the others are internal factors. Generally, external factors can indirectly affect consumers’ purchase willingness by affecting internal factors. For example, consumers are influenced by a range of external factors (such as taxation, financial subsidies, etc.) when they purchase energy-efficient equipment [45,46]. Some scholars caution that consumers may find themselves priced out of the market as green products are expensive [46]. Similarly, in Steg and Velk’s study, they claimed that external factors are likely to adjust the relationship between internal factors and behavior [47]. Therefore, hypotheses 7–14 were proposed based on interactions among these influencing factors, as follows:

**Hypothesis 7** **(H7).***Recycling product information has an indirect positive effect on contractors’ purchase willingness towards recycled C&DW products by influencing contractors’ attitude*.

**Hypothesis 8** **(H8).***Recycling product information has an indirect positive effect on contractors’ purchase willingness towards recycled C&DW products by influencing contractors’ subjective norm*.

**Hypothesis 9** **(H9).***Recycling product information has an indirect positive effect on contractors’ purchase willingness towards recycled C&DW products by influencing contractors’ perceived behavioral control*.

**Hypothesis 10** **(H10).***Recycling product information has an indirect positive effect on contractors’ purchase willingness towards recycled C&DW products by influencing contractors’ perceived consumer effectiveness*.

**Hypothesis 11** **(H11).***Government measures have an indirect positive effect on contractors’ purchase willingness towards recycled C&DW products by influencing contractors’ attitude*.

**Hypothesis 12** **(H12).***Government measures have an indirect positive effect on contractors’ purchase willingness towards recycled C&DW products by influencing contractors’ subjective norm*.

**Hypothesis 13** **(H13).***Government measures have an indirect positive effect on contractors’ purchase willingness towards recycled C&DW products by influencing contractors’ perceived behavioral control*.

**Hypothesis 14** **(H14).***Government measures have an indirect positive effect on contractors’ purchase willingness towards recycled C&DW products by influencing contractors’ perceived consumer effectiveness*.

In light of the hypotheses above and their interactions, an initial hypothetical model (see Figure 1) is accordingly developed.

## 3. Research Methods and Process

According to the general steps adopted in empirical study on influencing factors, this research involves the following four steps: (1) identifying potential influencing factors; (2) collecting data; (3) analyzing data; and (4) verifying and discussing determinants. Figure 2 explains the research process and methods. 

### 3.1. Identifying Potential Influencing Factors

Existing studies demonstrate that interviews, literature bibliometrics and focus group discussions are practical approaches to identifying potential influencing factors [21,22]. Generally, interviews and focus group discussions are performed to further complete and obtain potential influencing factors derived from the literature bibliometrics [21,26]. Compared with semi-structured interviews, coordinating with experts and arranging their time is more difficult when adopting focus group discussion. Thus, the former approach has gained wide popularity in research [21,26]. TPB lays the foundation for analyzing human’s attitude, willingness, and behavior and is widely accepted by scholars [19,24,26]. Therefore, this study took three steps to identify potential influencing factors. 

Firstly, three potential influencing factors affecting willingness were identified based on TPB. Next, other potential influencing factors were supplemented using the literature bibliometrics. Literature related to our research came from diverse databases such as Web of Science, ScienceDirect, etc., which could be found in Section 2. In the following step, five experts from the government, C&DW recycling enterprises, and contractors were invited to complete interviews to supplement these factors and confirm the factors found in the first two stages in November 2019. Finally, six factors influencing the contractors’ willingness to purchase recycled C&DW recycling products were identified according to the above two steps. Based on this, theoretical hypothesis and hypothetical model are established.

### 3.2. Collecting Quantitative Data

A questionnaire survey collect the sample data for empirical research through the acquisition of significant amounts of quantitative data [21]. This method is commonly used for studying stakeholders’ attitudes, willingness, and behaviors towards C&DW management [19,21,24,26,48]. Thus, a questionnaire survey is adopted by this study, which specifically includes the following steps:

#### 3.2.1. Preliminary Questionnaire

Normally, it is necessary to conduct preliminary surveys to ensure the validity of the questionnaire [19]. In this research, preliminary questionnaires were distributed to 10 respondents from three contractors in December 2019, including a chairman, two general managers, two department managers, and five project managers. Questions and suggestions they put forward have improved the readability and operability of the questionnaire. For instance, respondents suggested that the definition of recycling products and a five-point Likert scale scoring option before the measurement item corresponding to each variable should be added, etc. 

#### 3.2.2. Final Questionnaire

The final questionnaire contained two essential parts: (1) the demographic characteristics of respondents (e.g., gender, age, position, etc.) and information about their enterprises, viz. professional levels, grades, and registration types; (2) measurement items. Specifically, each measurement item contains five options: “strongly disagree”, “disagree”, “uncertain”, “agree”, or “strongly agree”, which respondents can choose according to their attitude (for the final questionnaire).

#### 3.2.3. Questionnaire Distribution and Collection

The Shenzhen Construction Industry Association, an organization consisting of 774 members (before 31 December 2019) and responsible for supervising and managing construction enterprises in Shenzhen, made a considerable contribution to distributing the questionnaires. Its annual conference was held on 9 January 2020. Typically, senior leaders (e.g., chairman, general manager) from all the members would attend this meeting, and their willingness to purchase recycled C&DW products favorably represents that of their enterprises. As the real decision-maker, the chairman or general manager is the perfect representative of the investigated enterprises [49]. Thus, it was an excellent opportunity to collect sample data for an empirical study. 

Questionnaires (500) were distributed by three rigorously trained graduate students from Shenzhen University, and two staff members from Shenzhen Construction Industry Association and 459 were finally returned. Among questionnaires received, 27 were incomplete, and 41 were filled out by respondents whose position level was not above the project manager. These 41 respondents’ willingness was not considered representative of their enterprises. Consequently, the number of valid questionnaires was 391, representing a response rate of 85.19%. The valid number of samples exceeded 200, outnumbering the minimum number required in the SEM [50].

#### 3.2.4. Respondents’ Information

The detailed demographic data and respondents’ enterprises are presented in Table 1. 

It can be found that the respondents were almost male (96.9%) and mainly concentrated between the ages of 35 to 55 (82.6%). The vast majority of respondents have worked for more than ten years (98%), and most of them received a bachelor’s or higher degree (93.1%). Besides, respondents were all middle- and senior-level leaders, including chairman (22.3%), general manager (33.0%), department manager (30.9%), and project manager (13.8%), and their willingness towards purchasing recycled C&DW products stands for that of their enterprises. The professional fields of the respondent’s enterprises were mainly housing construction engineering (32.5%), municipal public engineering (28.9%), and building decoration engineering (17.1%). These enterprises are the primary purchasers and users of recycled C&DW products. Hence, concerns about the contractors’ willingness to purchase the products can be realistically reflected through these respondents. 

### 3.3. Analyzing the Quantitative Data

The sample data collected in the previous stage needs a sequence of analysis (such as reliability analysis, confirmatory factor analysis, etc.) before finalizing determinants affecting contractors’ willingness towards purchasing recycled C&DW products. The questionnaire design’s reasonability correlates with the reliability of sample data, influencing the credibility of research results [21]. The reliability analysis is a necessary process to check the quality of the questionnaire. Only when the reliability requirement is met can the following research steps be taken correspondingly [20]. It should be noted that when the questionnaire contains multiple dimensions, it is necessary to test internal consistency for each dimension [26]. Cronbach’s alpha reliability is widely used to measure internal consistency reliability in previous studies [18,21,26]. Consequently, with the aid of SPSS 19, it was used to analyze the sample data’s reliability in this study. In the range from 0 to 1, the larger the alpha coefficient indicates, the better reliability among items. Normally, the value greater than or equal to 0.7 is considered high reliability [51,52]. 

In recent decades, SEM has been intensively utilized to carry out theoretical explorations and empirical research in various fields [53]. It stands out from other tools (e.g., factor analysis, multivariate regression, etc.) by simultaneously analyzing factors and paths [54]. Because of SEM’s superiority, researchers have made extensive use of it to investigate critical factors affecting stakeholders’ attitudes, willingness, and behaviors towards C&DW management [18,19,20]. Therefore, this study explores the determinants that affect contractors’ purchase willingness towards recycled C&DW products and analyze influencing factors and their interactions based on SEM.

For verifying the validity of the measurement model, a confirmatory factor should be performed to analyze the sample data before the SEM analysis [51]. Frequently, existing research selects the overall model fit to verify the degree of consistency between the presumptive model and actual data [21]. As shown in Table 2, the overall model fit includes three categories of goodness-of-fit criteria, each with several specific indicators to measure, five for absolute fit indices, five for incremental fit indices, and three for parsimonious fit indices [55]. Therefore, with the aid of the AMOS 21 platform (International Business Machines Corporation, New York, USA), this study conducts confirmatory factor analysis based on three index categories mentioned above. After confirmatory factor analysis, it needs to use the same indicators as above to further assess the overall model fit until the final optimized model is obtained [21]. It is imperative to compute estimated standardized path coefficients and their significance on this foundation, ultimately determining the critical influencing factors and its regression weights.

### 3.4. Verifying and Discussing Determinants

It is necessary to verify the results obtained from the SEM, analyze and discuss the reasons accounting for this result, to propose strategies better to promote contractor’s purchase willingness towards recycled C&DW products. Telephone interviews and face-to-face interviews are considered two efficient verification approaches [19]. This study invited three senior leaders from construction enterprises, including one chairman and two general managers, who all participated in the previous questionnaire survey. The interviewees have more than 20 years of work experience in the construction industry, and their enterprises have purchased recycled C&DW products in recent years. Face-to-face interviews were adopted on 19 and 20 January, 2020, as it facilitated the acquisition of respondents’ opinions and thoughts compared to telephone interviews. The interview content mainly covered the validation results for empirical study and discussion about the reason for this result.

## 4. Results

### 4.1. Reliability Analysis

As recommended by Wang et al., Cronbach’s alpha was utilized to measure the consistency reliability within the questionnaire based on average correlations among the items in this research [21]. The value greater than or equal to 0.7 indicates acceptable consistency, and the larger alpha coefficient indicates the better reliability among items [52]. In this study, the Cronbach’s alpha coefficient value of overall variables, and each latent variable exceeds 0.9 (see Table 3), which indicates that the sample data has excellent reliability.

### 4.2. Confirmatory Factor Analysis

Observation variables with factor load less than 0.5 should be deleted during the confirmatory factor analysis for the subsequent multiple regression analysis [19]. The full measurement model, which contains seven latent variables, is shown in Figure 3. Each latent variable is composed of several observed variables. No observation variables need to be excluded on account of no factor load of less than 0.5 in this model. Also, all the goodness-of-fit indices are within an acceptable range (see Table 4). It follows that the model and the data fit better, which means that the model is acceptable.

### 4.3. Multiple Regression Analysis

The initial structural model was obtained after the confirmatory factor analysis (see Figure 4). It contains seven first-order latent variables, each of which has several second-order observed variables to measure. The analysis results of the initial model (see Table 5) show that the sample data agrees with the initial model on adequate goodness-of-fit measures. 

Form Table 6, since the C.R. of path PW ← RPI and path SN ← RPI is less than 1.96 (*p* > 0.05), it shows that the interaction between the two paths in the initial model was not significant. Therefore, these insignificant paths should be removed. It is worth noting that the one with the highest *p*-value in the above two paths is unacceptable and should be dropped in the first place. The path from recycled product information (RPI) to purchase willingness (PW) was removed first to establish a new model based on the highest *p*-value of 0.447. It follows that H5 is invalid, i.e., recycling product information has no direct and remarkable impact on contractors’ willingness to purchase recycled C&DW products. 

After the first correction, it was found that the *p*-value and C.R. of the other paths except for the path from recycled product information (RPI) to the subjective norm (SN) met the requirements. A second correction was performed, viz. the path from recycling product information (RPI) to the subjective norm (SN) was deleted. This indicates that H8 is invalid.

After the second correction, all the remaining paths in the new model are satisfactory. Also, all goodness-of-fit indices are valid. This shows that the model obtained by the second correction is the final model (see Figure 4). The final model analysis results could also be found in Table 5. Combining the data on the final model in Table 6 and the preliminary hypothetical model (see Figure 1), the critical path of determinants affecting contractors’ purchase willingness towards recycled C&DW products is obtained (see Figure 5). It contains twelve key paths, which means that all twelve hypotheses except Hypothesis 5 and Hypothesis 8 have been verified. Table 7 shows the direct effect, indirect effect, and total effect of different influencing factors.

## 5. Discussions

### 5.1. The Critical Influencing Factors

Table 7 shows that the critical factors affecting contractors’ purchase willingness towards recycled C&DW products are government measures, purchase attitude, perceived behavioral control, and subjective norm, while perceived consumer effectiveness makes the least contributions. Although recycling product information has no direct positive effect, it indirectly affects contractors’ purchase willingness by influencing contractors’ purchase attitude, perceived behavioral control, and perceived consumer effectiveness.

#### 5.1.1. Government Measures

The empirical study has shown that government measures are the most critical influencing factor (the total effect is 0.48) among all factors. Despite the smallest direct effect (the direct effect is 0.16), they make a visibly indirect impact (the indirect effect is 0.32) on the contractors’ purchase willingness towards recycled C&DW products by influencing internal factors. Similarly, Zhao et al. found that government measures (referred to as financial subsidies and value tax credits in their research) are the main influencing factor for homeowners to adopt greener products in Florida [58]. Also, for influence path, they have an indirect effect on the purchase willingness. Additionally, economic incentive policy plays a more crucial part in increasing consumers’ purchase of low-carbon products than propaganda guidance policy in China [59]. In contrast, Wang et al. asserted that policies and media publicity do not form persuasion used to influence Chinese city dwellers’ intention to foot the bill for energy-efficient appliances [60]. These findings indicate that government measures result in different effects on increasing consumers’ willingness to purchase different types of environmentally friendly products, bringing policy-makers implications that policy measures must be formed according to market situations and take local conditions into considerations.

#### 5.1.2. Purchase Attitude

The results show that contractors’ willingness to purchase recycled C&DW products was also determined by their attitudes (the total effect is 0.33), which signifies the importance of a positive attitude towards recycled C&DW products among contractors while purchasing it. Coincidentally, attitudes also significantly affect low-carbon products’ purchase intention and energy-efficient equipment [46,59]. This result has also been well demonstrated in the field of C&DW research, e.g., attitude is considered the dominant motivational factor that determines project manager’s willingness towards C&DW reduction [26]. Besides, Wang et al. found that attitude is a prominent factor shaping designers’ willingness towards C&DW minimization [21]. These studies illustrate to no small extent that attitude is one of the determinants affecting willingness. Thus, for contractors, changing their attitudes towards recycled C&DW products could effectively increase their purchase willingness. 

#### 5.1.3. Perceived Behavioral Control

Perceived behavioral control, denoted as the convenience of purchasing recycled C&DW products for contractors, is also at play directly in improving the willingness to purchase recycled C&DW products (the total effect is 0.30). It is also a determinant shaping Chinese households’ willingness to recycle e-waste [29]. Similarly, perceived behavioral control has significantly influenced consumers’ willingness to purchase organic foods in India [61]. Unlike these findings, Li et al. claim that perceived behavioral control does not remarkably affect the purchase intention for low-carbon products, reflecting a mismatch in the determinants that affect consumers’ willingness to purchase different products also vary from area to area [59]. However, when it comes to recycled C&DW products, perceived behavioral control is one of the determinants affecting contractors’ purchase willingness in China.

#### 5.1.4. Subjective Norms

Subjective norms also positively affect contractors’ purchase willingness towards recycled C&DW products, which again validates the TPB proposed by Ajzen [27]. The finding echoes that in the previous research of Shuai et al., subjective norm exerts a substantial impact on the willingness to buy green products [62]. The research of Li et al. shows that consumers can gain information from the reference population around them who have low-carbon products’ purchasing experience [59]. Higher intention to purchase these goods is also attributable to the social atmosphere regarding buying these products. Their findings are well justified in this study, with more than 85% of respondents claiming that the atmosphere surrounding enterprises actively purchasing recycled C&DW products will increase their purchase willingness to a large extent. 

#### 5.1.5. Perceived Consumer Effectiveness

Data analysis proved that perceived consumer effectiveness is another crucial factor affecting contractors’ willingness to purchase recycled C&DW products. Nonetheless, it shows the minimum positive effect (the total effect is only 0.22), but it cannot be ignored. Contractors can achieve a sense of satisfaction by addressing environmental issues by switching to recycling products and ultimately enhancing their purchase willingness. Li et al. and Wang et al. have attested that perceived consumer effectiveness is a significant factor affecting consumers’ willingness to purchase low-carbon and energy-efficient products [59,60]. Currently, sustainable development continues to catch the limelight worldwide, and low-carbon consumption as a method of environmental protection has also been sought after by more and more people. These environmentally conscious consumers have endeavored to convert environmental consciousness into green consumption willingness. This trend provides a silver lining for China and even the world in the face of aggravating regional and global environmental degradation, whether it is related to the recycled C&DW products or the remaining green products. 

#### 5.1.6. Recycled Product Information

From the results, it can be found that, as an external factor, recycled product information exerts no significant direct impact on contractors’ willingness to purchase recycled C&DW products. Nevertheless, it indirectly affects contractors’ purchase willingness by influencing contractors’ purchase attitude, perceived behavioral control, and perceived consumer effectiveness (the indirect effect is 0.23). This aligns with Chang and Wildt’ findings, who revealed that product information is an essential factor affecting potential consumers for product evaluation and actual purchase [38]. The price of recycled C&DW products shows the most excellent effect on contractors’ purchase willingness comparing to the quality, brand, packaging, and sales channels. Generally, contractors tend to purchase recycled C&DW products if their price is lower and vice versa. Data analysis results are consistent with those of Annunziata and Vecchio that product manufacturers are also one of the main factors affecting consumers’ purchase willingness when choosing products [63]. This means that contractors will focus on the product manufacturer when purchasing recycled C&DW products.

### 5.2. Strategies for Increasing Purchase Willingness of Recycled C&DW Products

Strategies are presented separately from different perspectives of government, contractors, recycling enterprises, and public buyers based on empirical study and questionnaire survey results. 

#### 5.2.1. Strategies for Government

The empirical study results have turned out that H6, H11, H12, H13, and H14 are all supported, which means that the government takes the lead in increasing the purchase willingness of contractors towards recycled C&DW products. From the questionnaire survey results, the government’s incentive policy outperforms other government measures (e.g., launching demonstration project, increasing publicity) in encouraging contractors to make purchases of recycled C&DW products. One interviewee stated that financial subsidies and tax incentives were two effective measures to encourage contractors to purchase and use recycled C&DW products. This interviewee also agreed that the housing construction project could follow the policy of prefabricated buildings. When recycled products exceed a certain percentage, it could be rewarded by increasing the plot ratio. Besides, some interviewees suggested that using a certain percentage of recycled products is mandatory for applying for some awards. 

Besides, it is indispensable to strengthen the promotion of recycled C&DW products. For example, recycling enterprises regularly hold recycled product release conferences and introduce the latest recycled products to the public, such as product names, uses, and prices. The government can also increase the visibility and influence of recycled C&DW products by launching pilot projects and encouraging government-investment projects to purchase and use recycled products. The questionnaire survey results show that these measures would make contractors feel more confident and willing to buy. Also, government departments should formulate relevant norms and policies to ensure the quality of recycled products and stakeholders’ interests to promote the market’s standardized operation.

#### 5.2.2. Strategies for Contractors

H1, H2, H3, and H4 are all retained, indicating that contractor’s purchase willingness towards recycled C&DW products is mainly affected by their factors (e.g., purchase attitude, perceived behavioral control, subjective norm, and perceived consumer effectiveness). Many respondents have realized that substituting conventional materials with recycled C&DW products could address natural resource scarcity to a certain extent, which will help protect the environment and improve environmental quality. They also strongly agree that improper treatment of C&DW will do harm to human health and even cause safety accidents. The painful lesson of a catastrophic landslide caused by a C&DW landfill in Shenzhen, China, on 20 December 2015, makes the case for the danger of such inappropriate disposal stronger [64]. However, some respondents claim to be more willing to purchase and use non-recyclable products in the future because they know their purchasing channels, prices, performance, etc. More importantly, they have formed friendly and reliable cooperative relations with these production enterprises. Therefore, these contractors, they should try to change the inherent purchase model and re-examine recycled C&DW products. They should also be aware of energy conservation and environmental protection on the surface and practice it. 

Furthermore, contractors, who have already purchased and used recycled C&DW products, should increase publicity, encourage and recommend other contractors to purchase them since results derived from the questionnaire survey show that purchases willingness has a chain reaction (most of the respondents strongly agreed or agreed with the statement “If the surrounding enterprises purchase recycled C&DW products, my enterprise will be more willing to purchase”). Moreover, these contractors should regularly report recycled products and their needs to recycling enterprises to better facilitate the sustainable production and improvement of recycled C&DW products.

#### 5.2.3. Strategies for Recycling Enterprises

The results conducted from the questionnaire survey showed that it was difficult for some respondents to purchase recycled C&DW products. For instance, for the measurement items “I know exactly where to purchase recycled C&DW products”, “I know exactly the types of recycled C&DW products”, “I can easily inquire about recycled C&DW products”, and “I can easily purchase recycled C&DW products when I need them”, the results of these respondents were “uncertain”, “disagree”, and even “strongly disagree”. Therefore, recycling enterprises need to increase the promotion of recycled products. The establishment of recycled C&DW products trading and information inquiry platforms is considered an effective method to solve the above problems. Correspondingly, Shenzhen has developed and operated the “Excavated Soil and Rock Exchange and Recycled C&DW roduct Information Publishing Platform” since 2018 [65]. Recycling enterprises can regularly push the latest information about enterprises and recycling products through WeChat official accounts and web pages. 

The interviewees generally believe that reputable enterprises’ recycled products will help them to make purchase decisions. Thus, recycling enterprises should ensure product quality, provide high-quality after-sales service, and establish the right corporate image. Besides, like other commodities, price is an essential factor affecting contractors’ purchase willingness towards recycled C&DW products. Also, it is a bargaining chip to compete with other recycling enterprises. This means that recycling enterprises should increase investment in research and development of recycled products, focus on improving production technology and efficiency, and reduce production costs and sale prices.

#### 5.2.4. Strategies for Public Buyers

Changing the concept of the public is the key to promoting recycled C&DW products. Some public members have labelled them toxic, harmful, low quality, cheap, and have other misunderstandings about recycled products so that they are inclined to purchase non-recycled products at higher prices. As suggested by Yuan, to eliminate the public’s misconceptions towards recycled C&DW products and create a favorable social environment, awareness-raising campaigns related to energy conservation and emission reduction should be widely launched [66]. The public must also be deeply aware that purchasing and using recycled products can establish a personal energy conservation image. 

## 6. Conclusions

Expanding the purchase and use of recycled C&DW products is an effective means to promote environmentally sustainable development. This study investigated the key drivers which affect the contractor’s purchase willingness towards recycled C&DW products in Shenzhen, China. The initial hypothetical model was established based on the TPB. The questionnaire survey was then utilized in the data collection process. Influencing factors and their factorial interactions were further determined based on SEM. The empirical study results were finally analyzed and discussed through face-to-face interviews. 

Research results revealed that the contractor’s purchase willingness towards recycled C&DW products is affected by both internal and external contractor factors. Internal factors such as contractor’s attitude, subjective norms, perceived behavioral control, and perceived consumption effectiveness significantly and directly affect the contractor’s purchase willingness, and their degree of influence gradually decreases. External factors such as government measures and recycled product information indirectly affect the contractor’s purchase willingness by affecting internal factors. Correspondingly, strategies to increase the contractor’s purchase willingness towards recycled C&DW products are presented separately from different perspectives of government, contractors, recycling enterprises, and public buyers. These strategies include formulating incentive policies, increasing publicity, expanding sales channels, improving product quality, reducing production costs and sales prices, and changing traditional concepts.

This study focuses on the contractors’ willingness to purchase recycled C&DW products, which broadens the horizon of existing the C&DW management research fields. Besides, the empirical study’s findings can serve as a theoretical reference for government departments to develop promotion policies related to C&DW product recycling, which helps guide recycling enterprises to produce better and sell recycled C&DW products. 

There are some limitations to the study. For instance, this research mainly explored the determinants that affect contractors’ purchase willingness towards recycled C&DW products. Many uncertainties still exist in the relationship between willingness and behavior, for example, higher purchase willingness does not necessarily lead to more actual purchasing behavior. Therefore, it will be necessary to investigate the “willingness-behavior gap” in contractors and investigate the determinants that could transform willingness into actual purchasing behavior. 

## Figures and Tables

**Figure 1 ijerph-18-04412-f001:**
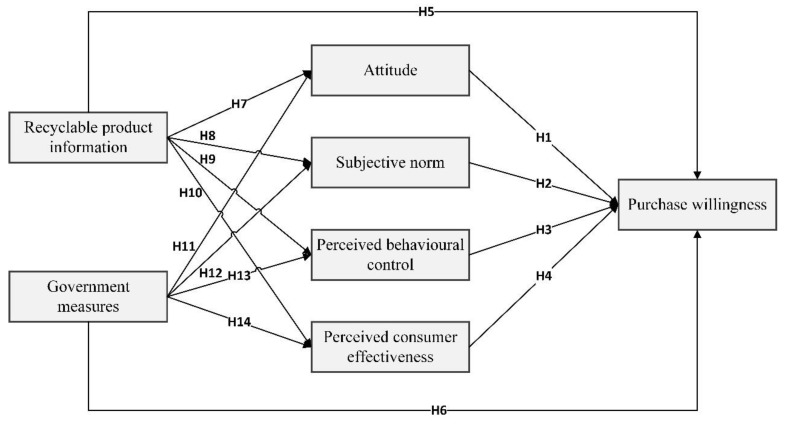
The preliminary hypothetical model.

**Figure 2 ijerph-18-04412-f002:**
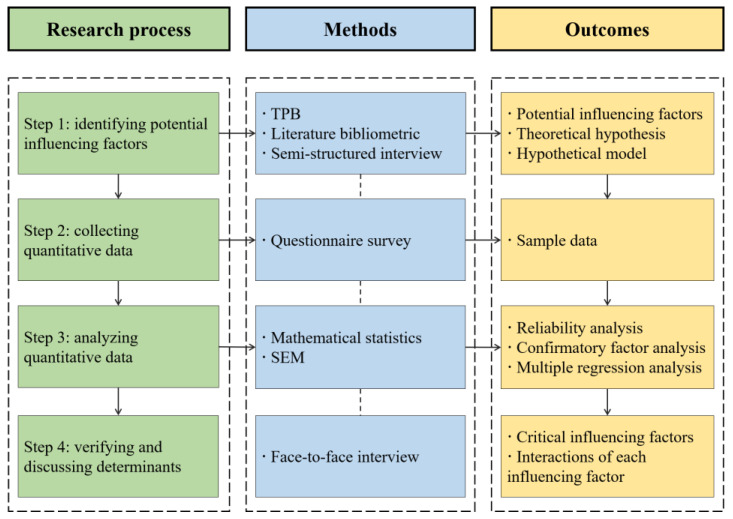
The research process.

**Figure 3 ijerph-18-04412-f003:**
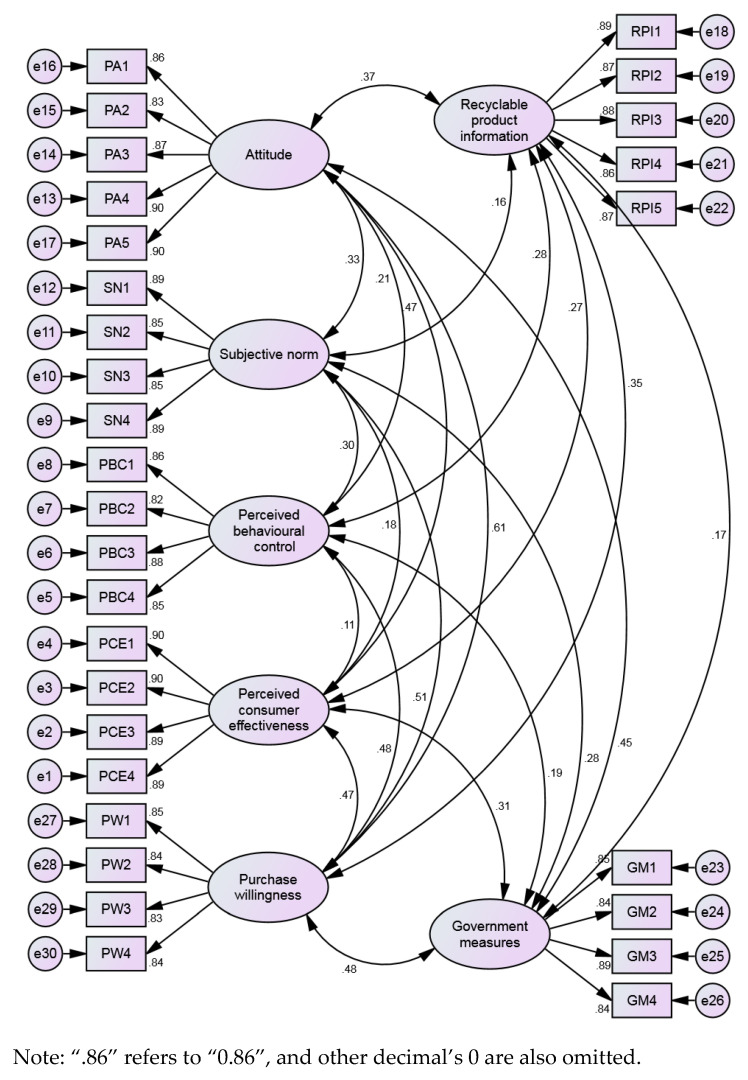
Standardized regression weights of the full measurement model.

**Figure 4 ijerph-18-04412-f004:**
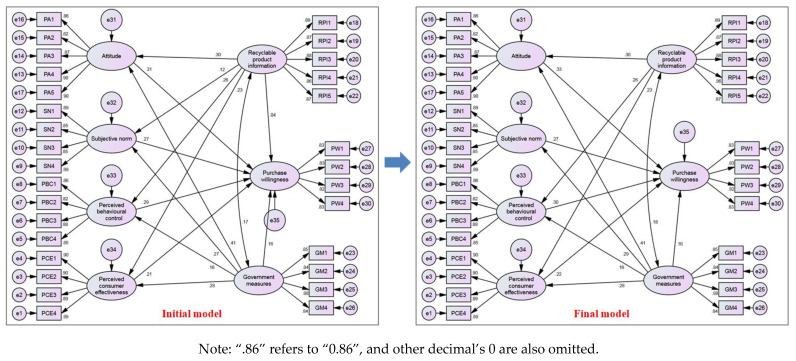
Standardized estimation of the initial and final models.

**Figure 5 ijerph-18-04412-f005:**
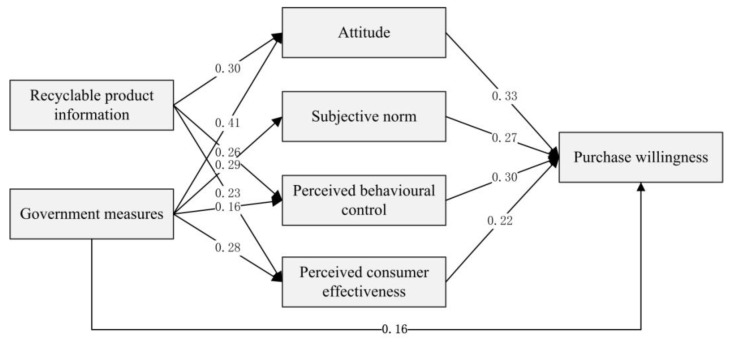
Key path of influencing factors.

**Table 1 ijerph-18-04412-t001:** Statistics of the respondents and their enterprises.

Variable		Category	Number	Percentage (%)
Personal information	Gender	Male	379	96.9
		Female	12	3.1
	Age (years old)	<35	13	3.3
		35–45	102	26.1
		45–55	221	56.5
		>55	55	14.1
	Work experience (years)	<10	8	2.0
		10–20	94	24.0
		20–30	200	51.2
		>30	89	22.8
	Educational level	Senior school or below	27	6.9
		Bachelor	272	69.6
		Master or above	92	23.5
	Position	Chairman	87	22.3
		General manager	129	33.0
		Department manager	121	30.9
		Project manager	54	13.8
Corporate Information	Professional field	Housing construction engineering	127	32.5
		Municipal public engineering	113	28.9
		Building decoration engineering	67	17.1
		Others ^a^	84	21.5
	Qualifications level	Premium	58	14.8
		Level 1	172	44.0
		Level 2	100	25.6
		Level 3	61	15.6
	Registration type	State-owned enterprise	147	37.6
		Private enterprise	215	55.0
		Others ^b^	29	7.4

^a^ “Others” refers to steel structure engineering, fire facility engineering, etc. ^b^ “Others” refers to Chinese-foreign joint ventures, foreign-funded enterprises, etc.

**Table 2 ijerph-18-04412-t002:** Indicators to verify the effectiveness of the model.

Overall Model Fit	Specific Indicators	Abbreviation
Absolute fit indices	Normalized chi-square	X^2^/df
	Root mean square residual	RMR
	Goodness-of-fit index	GFI
	Adjusted goodness-of-fit index	AGFI
	Root mean square error of approximation	RMSEA
Incremental fit indices	Normed fit index	NFI
	Comparative fit index	CFI
	Tucker–Lewis Index	TLI
	Incremental fit index	IFI
	Relative fit index	RFI
Parsimonious fit indices	Parsimony normed-fit index	PNFI
	Parsimony comparative fit index	PCFI
	Parsimony goodness-of-fit index	PGFI

**Table 3 ijerph-18-04412-t003:** Results of the reliability analysis.

Latent Variables	Cronbach’s Alpha	Overall Cronbach’s Alpha
Purchase willingness	0.903	0.923
Purchase attitude	0.937	
Subjective norm	0.926	
Perceived behavioral control	0.912	
Perceived consumer effectiveness	0.940	
Recycled product information	0.941	
Government measures	0.918	

**Table 4 ijerph-18-04412-t004:** Goodness-of-fit of the initial measurement model.

Goodness-of-Fit Measure		Level of Acceptance Fit ^a^	Fit Statistics
Absolute fit	X^2^/df	<5 (preferably 1–2)	1.210
	RMR	<0.05	0.034
	GFI	0–1 (no fit–perfect fit)	0.893
	AGFI	0–1 (no fit–perfect fit)	0.871
	RMSEA	<0.10 (preferably <0.08)	0.028
Incremental fit	NFI	0–1 (no fit–perfect fit)	0.934
	IFI	0–1 (no fit–perfect fit)	0.988
	CFI	0–1 (no fit–perfect fit)	0.988
	RFI	0–1 (no fit–perfect fit)	0.925
	TLI	0–1 (no fit–perfect fit)	0.986
Parsimonious fit	PNFI	>0.5	0.824
	PGFI	>0.5	0.738
	PCFI	>0.5	0.872

^a^ Thresholds are adapted from Xiong et al. [55], Ajayi and Oyedele [56], and Chen et al. [57].

**Table 5 ijerph-18-04412-t005:** Goodness-of-fit of the model.

Goodness-of-Fit Measure		Level of Acceptance Fit ^a^	Fit Statistics	
			Initial Model	Final Model
Absolute fit	X^2^/df	<5 (preferably 1–2)	1.319	1.323
	RMR	<0.05	0.074	0.083
	GFI	0–1 (no fit–perfect fit)	0.882	0.881
	AGFI	0–1 (no fit–perfect fit)	0.859	0.859
	RMSEA	<0.10 (preferably <0.08)	0.035	0.035
Incremental fit	NFI	0–1 (no fit–perfect fit)	0.927	0.926
	IFI	0–1 (no fit–perfect fit)	0.981	0.981
	CFI	0–1 (no fit–perfect fit)	0.981	0.981
	RFI	0–1 (no fit–perfect fit)	0.918	0.918
	TLI	0–1 (no fit–perfect fit)	0.979	0.979
Parsimonious fit	PNFI	>0.5	0.831	0.835
	PGFI	>0.5	0.740	0.743
	PCFI	>0.5	0.880	0.884

^a^ Thresholds are adapted from Xiong et al. [55], Ajayi and Oyedele [56], and Chen et al. [57].

**Table 6 ijerph-18-04412-t006:** Regression weights in the model.

Path ^a^	Estimate		S.E.		C.R.		P	
	Initial Model	Final Model	Initial Model	Final Model	Initial Model	Final Model	Initial Model	Final Model
PW ← PA	0.309	0.325	0.048	0.046	5.171	5.693	*** ^c^	***
PW ← SN	0.265	0.267	0.044	0.043	5.081	5.101	***	***
PW ← PBC	0.289	0.302	0.046	0.045	5.417	5.768	***	***
PW ← PCE	0.213	0.222	0.041	0.040	4.031	4.245	***	***
PW ← RPI	0.042	/ ^b^	0.046	/	0.761	/	0.447	/
PW ← GM	0.161	0.159	0.055	0.055	2.655	2.601	0.008	0.009
PA ← RPI	0.305	0.303	0.060	0.060	5.194	5.158	***	***
SN ← RPI	0.125	/	0.064	/	1.929	/	0.054	/
PBC ← RPI	0.259	0.258	0.062	0.062	3.934	3.922	***	***
PCE ← RPI	0.226	0.225	0.067	0.067	3.591	3.573	***	***
PA ← GM	0.407	0.407	0.068	0.068	6.672	6.671	***	***
SN ← GM	0.267	0.291	0.072	0.072	4.017	4.395	***	***
PBC ← GM	0.158	0.158	0.069	0.069	2.389	2.391	0.017	0.017
PCE ← GM	0.283	0.283	0.076	0.076	4.385	4.383	***	***

^a^ PW = Purchase willingness, PA = Purchase attitude, SN = Subjective norm, PBC = Perceived behavioral control, PCE = Perceived consumer effectiveness, RPI = Recycled product information, GM = Government measures. ^b^ The path has been deleted. ^c^ Statistically significant at the 0.001 level of confidence.

**Table 7 ijerph-18-04412-t007:** The direct effect, indirect effect and total effect of predictors.

Variables	PA	SN	PBC	PCE	RPI	GM
Direct effect	0.33	0.27	0.30	0.22	-	0.16
Indirect effect	-	-	-	-	0.23	0.32
Total effect	0.33	0.27	0.30	0.22	0.23	0.48

## Data Availability

We have participated sufficiently in work to take public responsibility for the appropriateness of the collection, analysis, and interpretation of the data.

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
