# Peer review of "Determinants Affecting Purchase Willingness of Contractors towards Construction and Demolition Waste Recycling Products: An Empirical Study in Shenzhen, China"

_ijerph, 2021, doi:10.3390/ijerph18094412_

Round 1
Reviewer 1 Report
The manuscript entitled "Determinants Affecting Purchase Willingness of Contractors towards Construction and Demolition Waste Recycling Products: An Empirical Study in Shenzhen, China" presents a method of successfully addressing construction and demolition waste environmental problems. This study explored the key drivers using the Exploratory Sequential Mixed Approach, developing a hypothetical model based on the theory of planned behavior along with survey data. This is a comprehensive paper involving extensive research. The manuscript is well-structured and presented. The goals and motivation of the study are quite clear. The manuscript meets the requirements of the International Journal of Environmental Research and Public Health.
Additional comments that need to be addressed:
1) A minor number of typographical/grammatical issues need to be dealt with, for example
The sentence “Form Table 6, the interaction between the two paths in the initial model was not significant because their C.R. was less than 1.96 (P>0.05) should be “From Table 6, the interaction between the two paths in the initial model was not significant because their C.R. was less than 1.96 (P>0.05)” should be “From Table 6, the interaction between the two paths in the initial model was not significant because their C.R. was less than 1.96 (P>0.05) should be “From Table 6, the interaction between the two paths in the initial model was not significant because their C.R. was less than 1.96 (P>0.05)”
2) The references are presented in an unacceptable manner. They are not numbered and cannot be found in the manuscript.
Author Response
1. A minor number of typographical/ grammatical issues need to be dealt with, for example, the sentence “Form Table 6, the interaction between the two paths in the initial model was not significant because their C.R. was less than 1.96 (P>0.05)”.
Thanks for your reminder. We have carefully reviewed the manuscript and modified it. For example, the sentence “Form Table 6, the interaction between the two paths in the initial model was not significant because their C.R. was less than 1.96 (P>0.05)” has been modified to “Form Table 6, since the C.R. of path PW ß RPI and path SN ß RPI is less than 1.96 (P>0.05), it shows that the interaction between the two paths in the initial model was not significant.”
2. The references are presented in an unacceptable manner. They are not numbered and cannot be found in the manuscript.
Pretty sorry for our reference format does not match. We have carefully read the author's guidelines and other latest published papers in International Journal of Environmental Research and Public Health, and we have re-edited the whole reference format.
Reviewer 2 Report
The manuscript is interesting, as it tries to investigate the determinants affecting contractors’ purchase willingness towards construction and demolition waste recycling products. Moreover, it analyzes and presents well the interactions among the influencing factors affecting contractors’ purchase willingness towards C&DW recycling products. Through its findings, it tries to propose a series of strategies to promote the purchase of the same products for various stakeholders (the government, contractors, recycling enterprises).
Specific comments and remarks:
Please follow the journal guideline. Please check references and style of citing literature.
Line 368 Figure 3 & Figure 5 -Accurate description of figure should be added.
Section 3.2. Do Authors believe that the sample selected for the questionnaire survey creates some biases?
Many interesting results arise from the survey and they are presented detailly. However, authors may add a summary of them in the Discussion sector.
Author Response
1. Please follow the journal guideline. Please check references and style of citing literature.
Thanks for your invaluable comment. In the revised manuscript, we have carefully read the author's guidelines and we have re-edited the whole reference format.
2. Line 368 Figure 3 & Figure 5 -Accurate description of figure should be added.
We appreciate your comments. In the revised manuscript, we have added the descriptions about Figure 3 and Figure 5, as follows:
“Observation variables with factor load less than 0.5 should be deleted during the confirmatory factor analysis for the subsequent multiple regression analysis [19]. The full measurement model is shown in Fig. 3, which contains seven latent variables. Each latent variable is composed of several observed variables. No observation variables need to be excluded on account of no factor load of less than 0.5 in this model. Also, all the good-ness-of-fit indices are within an acceptable range (see Table 4). It follows that the model and the data fit better, which means that the model is acceptable.”
“After the second correction, all the remaining paths in the new model are satisfactory. Also, all goodness-of-fit indices are valid. This shows that the model obtained by the second correction is the final model (see Fig. 4). The final model analysis results could also be found in Table 5. Combining the data on the final model in Table 6 and the preliminary hypothetical model (see Fig. 1), the critical path of determinants affecting contractors’ purchase willingness towards C&DW recycling products is obtained (see Fig. 5). It contains twelve key paths, which means that all twelve hypotheses except Hypothesis 5 and Hypothesis 8 have been verified. Table 7 shows the direct effect, indirect effect, and total effect of different influencing factors.”
3. Section 3.2. Do Authors believe that the sample selected for the questionnaire survey creates some biases?
Thank you for your valuable question. This study aims to explore the determinants affecting contractors’ purchase willingness towards C&DW recycling products. Therefore, the contractor is the interviewee in this study. In addition, the chairman or general manager is usually the real decision-maker, and their willingness towards purchasing C&DW recycling products favorably stands for that of their enterprises. Therefore, the interviewees in this study are all leaders above the project manager level. We believe that the interviewees selected in this study are reasonable and the results obtained are also valid. The above related content could be found in the latest manuscript, as follows:
“Shenzhen Construction Industry Association, an organization consisting of 774 members (before December 31, 2019) and responsible for supervising and managing con-struction enterprises in Shenzhen, made a considerable contribution to distributing ques-tionnaires. Its annual conference was held on January 9, 2020. Typically, senior leaders (e.g. chairman, general manager) from all the members would attend this meeting, and their willingness towards purchasing C&DW recycling products favorably stands for that of their enterprises. As the real decision-maker, the chairman or general manager is the perfect representative of the investigated enterprises [49]. Thus, it was an excellent opportunity to collect sample data for an empirical study.”
4. Many interesting results arise from the survey and they are presented detailly. However, authors may add a summary of them in the Discussion sector.
Thanks for your invaluable comments. In the 5- Discussion part of revised manuscript, we first elaborated and discussed the results of the six influencing factors, and they were listed in the form of titles. On this basis, this manuscript also presented strategies from different perspectives of government, contractors, recycling enterprises, and public buyers for increasing purchase willingness of C&DW recycling products. Strategies from different perspectives are also listed in the form of titles. In addition, a brief summary is added before each part of the content. The structure of the 5- Discussion part is as follows:
- Discussions
5.1. The critical influencing factors
5.1.1. Government measures
5.1.2. Purchase attitude
5.1.3. Perceived behavioral control
5.1.4. Subjective norm
5.1.5. Perceived consumer effectiveness
5.1.6. Recycling product information
5.2. Strategies for increasing purchase willingness of C&DW recycling products
5.2.1. Strategies for government
5.2.2. Strategies for contractors
5.2.3. Strategies for recycling enterprises
5.2.4. Strategies for public buyers
Reviewer 3 Report
This study aims to explore the determinants affecting contractors’ purchase willingness towards C&DW recycling products and analyze their interactions which, in my opinion, it is a very interesting work from the point of view of the use of recycled products and waste from civil construction.
The developed work investigates key drivers that affect the contractor’s purchase willingness towards C&DW recycling products. The methodology used consist in a hypothetical model based on TPB, a survey and a structural equation modeling (SEM) for data analysis. The SEM is adequate for the analyses carried out, namely the influencing factors studied as most of them are related with behaviors, perceptions, attitudes, and fillings.
The paper has a very well-organized structure and with a very logical sequence.
The results show that several factors directly affect the contractor's willingness to buy in relation to the recycling of C&DW products. Based on the findings, are proposed some strategies to promote the purchase of C&DW recycling products for various stakeholders, such as the government, contractors, recycling enterprises, and public buyers.
As mentioned in the conclusions of the paper, the empirical study’s findings can serve as a theoretical reference for government departments to develop promotion policies related to the C&DW recycling products, which helps guide recycling enterprises to produce better and sell C&DW recycling products.
I consider that the work carried out brings important data and information for future work and that the continuity of the study will be very important to assess the uncertainties that still exist, namely in the relationship between will and behavior, as mentioned as mentioned in the paper conclusions.
Author Response
Pretty thank you for your acknowledge for our contributions in this paper. Also, thank you very much for reviewing our manuscript.